



# Technical note: Absorption aerosol optical depth components from AERONET observations of mixed dust plumes

Sung-Kyun Shin[1], Matthias Tesche[1], Detlef Müller[1], and Youngmin Noh[2]

[1]School of Physics, Astronomy and Mathematics, University of Hertfordshire, Hatfield, Hertfordshire, United Kingdom
[2]Department of Environmental Engineering, Pukyong National University, Busan, Republic of Korea

**Correspondence:** Youngmin Noh (nym1120@gmail.com)

**Abstract.** Absorption aerosol optical depth (AAOD) as obtained from sun/sky photometer measurements provides a measure of the light-absorbing properties of the columnar aerosol loading. However, it is not an unambiguous, aerosol-type specific parameter, particularly if several types of absorbing aerosols, for instance black carbon (BC) and mineral dust, are present in a mixed aerosol plume. The contribution of mineral dust to total aerosol light-absorption is particularly important at UV wavelengths. In this study we refine a lidar-based technique for the separation of dust and non-dust aerosol types for the use with Aerosol Robotic Network (AERONET) direct sun and inversion products. We extend the methodology to retrieve AAOD related to non-dust aerosol ($AAOD_{nd}$) and BC ($AAOD_{BC}$). We test the method at selected AERONET sites that are frequently affected by aerosol plumes that contain a mixture of Saharan or Asian mineral dust and biomass-burning smoke or anthropogenic pollution, respectively. We find that aerosol optical depth (AOD) related to mineral dust as obtained with our methodology is frequently smaller than coarse-mode AOD. This suggests that the latter is not an ideal proxy for estimating the contribution of mineral dust to mixed dust plumes. We present the results of the $AAOD_{BC}$ retrieval for the selected AERONET sites and compare them to coincident values provided in the Copernicus Atmospheric Monitoring System aerosol re-analysis. We find that modelled and AERONET $AAOD_{BC}$ are most consistent for Asian sites or at Saharan sites with strong local anthropogenic sources.

## 1 Introduction

Atmospheric aerosols have a strong impact on the Earth's radiation budget and climate (*Stocker et al.*, 2013). The main interactions between atmospheric particles and the climate system are through scattering and absorption of radiation (direct effect) and through modification of the microphysical properties of clouds (indirect effect). Estimates of the aerosol radiative forcing, i.e. of the perturbation of radiant fluxes due to aerosol particles, require information on aerosol loading as well as on the aerosol's optical and microphysical properties (*Bellouin et al.*, 2013). Aerosol optical depth (AOD) is the height integral of the aerosol extinction coefficient. It provides a measure of the columnar aerosol loading and is routinely obtained from ground-based and spaceborne remote-sensing observations. Despite our unprecedented global coverage of atmospheric aerosol information, it is still challenging to assess the aerosol radiative effect accurately. Not only are the sources of aerosols, their lifetime and the processes that affect their optical and microphysical characteristics highly inhomogeneous in space and time (*Stocker et al.*,



2013). Aerosol particles from different natural and anthropogenic sources also often mix and undergo aging processes, which reflects in the optical and microphysical properties of the bulk aerosol. Better estimates of the aerosol radiative forcing require an improved consideration of the properties and contributions of the different aerosol types in mixed aerosol plumes.

Remote sensing measurements are an important way to obtain insight into optical and microphysical aerosol properties. For instance, ground-based AErosol RObotic NETwork (AERONET, *Holben et al.* 1998, 2001) sun/sky radiometers provide long-term observations of aerosol products including spectral AOD, particle size distribution, and complex refractive index for the atmospheric column even at remote locations. AERONET also provides absorption aerosol optical depth (AAOD) which is a measure of the column aerosol loading of light-absorbing particles such as black carbon (BC), carbonaceous aerosols or mineral dust. However, AAOD becomes ambiguous if several types of absorbing aerosols are present in a mixed aerosol plume.

In dust-free conditions, BC as emitted from incomplete anthropogenic combustion or biomass burning is generally considered the main light absorber among atmospheric aerosols (*Bond and Bergstrom*, 2006; *Bond et al.*, 2013; *Russell et al.*, 2010), and thus, the main contributor to non-dust AAOD. *Schuster et al.* (2005) inferred columnar BC concentrations based on the Maxwell Garnett effective medium approximation with AERONET-retrieved complex refractive indices. *Koven and Fung* (2006) separated the absorption properties of BC from the absorption of dust by exploiting the spectral absorption properties inferred from the AERONET inversion. *Russell et al.* (2010) utilized AERONET-retrieved SSA, AAOD, and absorption Ångström exponent (AAE) as indicator to separate the contributions of BC, organic matter (OM), and mineral dust to the absorbing aerosol fraction.

Passive remote-sensing techniques can only provide the properties of the total aerosol mixture. Determining the optical properties of a certain aerosol type in a mixed aerosol plume requires additional information. For instance, the Ångström exponent (AE or *å*, *Ångström* 1964) as inferred from spectral AOD measurements gives qualitative information on aerosol size that can be used for aerosol-type classification and to infer the fine or coarse mode fraction in the aerosol size distribution (*Schuster et al.*, 2006). More detailed and quantitative information can be obtained from active aerosol remote sensing with lidar. In particular, the particle linear depolarization ratio (PLDR or $\delta$) is an intensive parameter that is very sensitive to particle shape. It can be used to obtain the contribution of dust and non-dust particles to a mixed aerosol plume under the assumption that this plume consists of only those two aerosol types in an external mixture (*Shimizu et al.*, 2004; *Tesche et al.*, 2009b). *Burton et al.* (2014) developed a generalised version of the methodology to separate contributions to mixtures of two aerosol types while *Mamouri and Ansmann* (2014) further refined it to also separate between the contribution of fine and coarse dust particles.

In this study, we use AERONET version 3 level 2 products to refine the lidar-based aerosol-type separation methodology to resolve the contributions of dust and non-dust aerosol to the total and absorbing fractions of AOD. This is most useful over and downwind of deserts where mineral dust can contribute significantly to AAOD – particularly at short wavelengths. We also propose a method to obtain the fraction of BC-related absorption to the non-dust AAOD. We describe our methodology in Section 2. In section 3, we present and discuss our results. We summarise our findings and provide concluding remarks in Section 4.





## 2 Data and methodology

### 2.1 AERONET sun/sky radiometer observations

AERONET (http://aeronet.gsfc.gov, *Holben et al.* 1998, 2001) operates automatic sun/sky radiometers for direct sun and sky radiation observation at sites all over the globe. AERONET instruments measure AOD at several wavelengths from 340 nm to
1640 nm. The AOD uncertainty is estimated as 0.01 to 0.02 depending on wavelength in the absence of cloud contamination. The calibrated sky radiance measurements typically have uncertainties below 5%. The Ångström exponent and the fine-mode fraction (FMF, *O'Neill et al.* 2003) are obtained from the spectral AOD measurements. The AERONET inversion is performed for measurements with a 440-nm AOD larger than 0.4 (*Dubovik et al.*, 2006). It uses direct-sun and sky-radiance measurements at 440, 675, 870, and 1020 nm to infer columnar particle properties such as the volume size distribution, the complex refractive
index, and the single-scattering albedo (SSA or $\omega$). The uncertainty in SSA is expected to be of the order of 0.03 (*Holben et al.*, 1998). Knowledge of SSA allows to determine the fraction of AOD related to light absorption, referred to as absorption aerosol optical depth (AAOD) as:

$$AAOD = (1 - \omega) \times AOD. \tag{1}$$

Detailed descriptions of the instrumentation, calibration, methodology, data processing, and data quality assurance are pro-
vided in *Holben et al.* (1998, 2001), *Dubovik et al.* (2002, 2006), *Eck et al.* (2005) and *Giles et al.* (2018). The recently released version 3 of the AERONET aerosol retrieval added spectral PLDRs and lidar ratios ($S$) to the list of inversion products. The representativeness of these values for pure mineral dust conditions has recently been discussed by *Shin et al.* (2018). In this contribution we use AERONET version 3 level 2.0 inversion products inferred from observations of mineral dust downwind of the Saharan and Asian deserts.

### 2.2 AOD and AAOD components in mixed dust plumes

In order to retrieve the AOD and AAOD for non-dust aerosols in mixed dust plumes, the optical properties of the mixture need to be separated according to the contributions of dust and non-dust particles, respectively. This is possible by using lidar measurements of the PLDR $\delta$ which depends mainly on the shape of the particles and their size with respect to the measurement wavelength. The PLDR is zero for spheres and increases with increasing particle non-sphericity. *Tesche et al.* (2009b)
present a method to separate mixtures of Saharan dust and biomass burning particles while *Shimizu et al.* (2004) retrieved the contribution of dust and non-dust particles in plumes of Asian dust mixed with spherical particles. *Noh* (2014) expanded these methods to retrieve the fractional contribution of the different aerosol types in the mixture to the bulk measurements of SSA, as well as the SSA for dust ($\omega_{\mathrm{d}}$) and non-dust ($\omega_{\mathrm{nd}}$) particles.

While $\delta$ is measured directly with lidar, it can also be computed from AERONET data and has been included as a standard
product in version 3 of the AERONET retrieval. For an external aerosol mixture, it is used to calculate the contribution of dust



($R_\mathrm{d}$) and non-dust ($R_\mathrm{nd}$) to the particle backscatter coefficient following *Shimizu et al.* (2004); *Tesche et al.* (2009b) as:

$$R_\mathrm{d} = \frac{(\delta - \delta_\mathrm{nd})(1 + \delta_\mathrm{d})}{(\delta_\mathrm{d} - \delta_\mathrm{nd})(1 + \delta)} \tag{2}$$

and

$$R_{\mathrm{n}d} = 1 - R_\mathrm{d}. \tag{3}$$

Here, $\delta_\mathrm{d}$ and $\delta_\mathrm{nd}$ indicate $\delta$ of dust and non-dust particles, respectively. Their values can be determined from lidar measurements (*Burton et al.*, 2014; *Freudenthaler et al.*, 2009) or from AERONET observations representative for pure mineral dust (*Shin et al.*, 2018). At the standard lidar wavelength of 532 nm, typical values are $\delta_\mathrm{d} = 0.33$ and $\delta_\mathrm{nd} = 0.02$ (*Freudenthaler et al.*, 2009; *Burton et al.*, 2014). *Shin et al.* (2018) recently discussed AERONET-derived $\delta_\mathrm{d}$ for mineral dust from different source regions. They conclude that in general, values of $\delta$ at 870 and 1020 nm from the AERONET version 3 inversion product seem

to be most reliable when compared to the literature on lidar observations of mineral dust. We consequently apply the aerosol-type separation procedure to AERONET measurements at 1020 nm using values of $\delta_\mathrm{nd} = 0.02$ and $\delta_\mathrm{d} = 0.30$ ($\delta_\mathrm{d} = 0.31$ for mixed Asian (Saharan) dust plumes (*Shin et al.*, 2018). When $\delta$ was lower than $\delta_\mathrm{nd}$ or higher than $\delta_\mathrm{d}$ , $R_\mathrm{d}$ was set to 0 or 1, respectively.

      The ratios $R_\mathrm{d}$ and $R_\mathrm{nd}$ obtained from using $\delta$ refer to the lidar measurements in the backscatter direction (i.e. the scattering

angle of 180°) and allow for inferring the dust-related backscatter coefficient $\beta_\mathrm{d}$ as:

$$\beta_\mathrm{d} = \beta R_\mathrm{d}. \tag{4}$$

      This approach needs to be refined so that it can be also applied to sun/sky photometer measurements which provide information on total light extinction, i.e. AOD is the height integral of the extinction coefficient $\alpha$, rather than the backscatter coefficient. For a single aerosol layer of depth $h$, it can be expressed as $AOD = \alpha h$. The extinction coefficient is connected to

$\beta$ through the lidar ratio $S = \alpha/\beta$. Consequently, dust AOD can be expressed as:

$$AOD_\mathrm{d} = S_\mathrm{d} \beta_\mathrm{d} h. \tag{5}$$

      The use of Eq. (5) for the total aerosol and the dust fraction together with Eq. (4) leads to the dust and non-dust AOD as:

$$AOD_\mathrm{d} = AOD \times R_\mathrm{d} \times \frac{S_\mathrm{d}}{S} \tag{6}$$

and

$$AOD_\mathrm{nd} = AOD - AOD_\mathrm{d}. \tag{7}$$

      AOD and $S$ are the total AOD and lidar ratio of the aerosol mixture as provided by AERONET, respectively. The $S_\mathrm{d}$ is the AERONET-derived lidar ratio of pure dust particles. It varies according to the desert source. We take the values of 44 sr and 54 sr for Asian and Saharan dust, respectively from *Shin et al.* (2018). As before, values at 1020 nm are used in the calculation.





To convert the 1020-nm AOD to other wavelengths $\lambda$, we use the Ångström exponent $\mathring{a}_d = 0.06 \pm 0.21$ for pure Saharan dust (*Tesche et al.*, 2009a). We obtain:

$$AOD_{d,\lambda} = AOD_{d,1020} \times \left(\frac{1020\,nm}{\lambda}\right)^{\mathring{a}_d} \tag{8}$$

and

$$AOD_{nd,\lambda} = AOD_\lambda - AOD_{d,\lambda} . \tag{9}$$

The contributions of dust and non-dust aerosols to the total AOD can now be described by the extinction-related dust ratio $\chi$ as:

$$\chi_{d,\lambda} = \frac{AOD_{d,\lambda}}{AOD_\lambda} = R_d \frac{S_d}{S} \tag{10}$$

and

$$\chi_{nd,\lambda} = \frac{AOD_{nd,\lambda}}{AOD_\lambda} = 1 - R_d \frac{S_d}{S} . \tag{11}$$

This means that the contribution of mineral dust to the extinction coefficient decreases (increases) with respect to the contribution to the backscatter coefficient (i.e. to $R_d$) if the second aerosol type in the mixture has a lidar ratio larger (smaller) than that of mineral dust. Mixtures with absorbing aerosols will show total lidar ratios larger than that of pure dust, which means that in the cases considered here, $\chi_d$ is generally smaller than $R_d$. The total SSA of the mixed dust/pollution plume as provided by individual AERONET measurements is now considered to be the result of mixing the SSA of dust and non-dust particles following the mixing rule:

$$\omega_\lambda = \chi_{d,\lambda}\omega_{d,\lambda} + \chi_{nd,\lambda}\omega_{nd,\lambda} . \tag{12}$$

Re-arranging Eq. (11) gives the SSA related to non-dust particles

$$\omega_{nd,\lambda} = \frac{\omega_\lambda - \chi_{d,\lambda}\omega_{d,\lambda}}{\chi_{nd,\lambda}} . \tag{13}$$

The spectral SSA for pure dust particles is taken from the literature (see Table 1). The non-dust fraction to AAOD can now be derived as

$$AAOD_{nd,\lambda} = (1 - \omega_{nd,\lambda})AOD_{nd,\lambda} . \tag{14}$$

We can assume that the light-absorbing features of the non-dust part of the aerosol plume are caused primarily by BC. As it has been shown that BC is not an ideal light absorber, i.e., $\omega_{BC,\lambda} \neq 0$, (*Bond and Bergstrom*, 2006; *Bond et al.*, 2013), we need to account for the SSA of BC to obtain the BC-related AAOD as:

$$AAOD_{BC,\lambda} = AOD_{nd,\lambda}(1 - \omega_{nd,\lambda})(1 - \omega_{BC,\lambda}) = AAOD_{nd,\lambda}(1 - \omega_{BC,\lambda}) . \tag{15}$$

*Bond and Bergstrom* (2006) report on the single-scattering albedo of 0.10 to 0.28 for fresh BC. Similar values for fresh BC have also been reported by *Khalizov et al.* (2009) and *Cross et al.* (2010). Here, we use values of $\omega_{BC,\lambda}$ from *Haywood and Ramaswamy* (1998). They are provided together with the other input parameters in Table 1.





### 2.3 Connection between $AAOD$, $AAOD_\mathrm{nd}$ and $AAOD_\mathrm{BC}$

Substituting Eq. (12) in Eq. (1) leads to the equation for the AAOD of dusty mixtures that accounts for the contribution of the different components as:

$$AAOD = (1 - (\chi_{\mathrm{d},\lambda}\omega_{\mathrm{d},\lambda} + \chi_{\mathrm{nd},\lambda}\omega_{\mathrm{nd},\lambda}))AOD\,. \tag{16}$$

The connection between total and non-dust AAOD for non-dust components with different values of $\omega_\mathrm{nd}$ between 0.90 and 0.96, an $\omega_\mathrm{d}$ of 0.98, and a total AOD of unity is presented in Figure 1. In case of $\chi_\mathrm{d} = 1$, all absorption is due to mineral dust. As the contribution of dust to the mixture decreases, the overall AAOD increases as a result of the stronger absorption of the non-dust particles. The ratio between $AAOD_\mathrm{nd}$ and total AAOD in Figure 1 changes linearly with $\chi_\mathrm{d}$ in case of equal values of $\omega_\mathrm{nd}$ and $\omega_\mathrm{d}$. The relation becomes increasingly non-linear with increasing difference in the absorbing properties of the dust

and non-dust particles. This means that total AAOD as provided by AERONET for dusty mixtures is likely to represent the non-dust component at larger wavelengths, where dust is less absorbing, while its interpretation is less ambiguous at shorter wavelengths.

The approach described above assumes that BC is the major absorber in mixtures of non-dust aerosols. Because $\omega_\mathrm{BC}$ is not zero, it is obvious from Eq. (15) that $AAOD_\mathrm{BC}$ is always smaller than AAOD and vanishes as $AAOD_\mathrm{nd}$ disappears, i.e. for

$\chi_\mathrm{d} = 1$.

### 2.4 CAMS aerosol re-analysis

We use the European Centre for Medium-range Weather Forecast's (ECMWF) Copernicus Atmospheric Monitoring Service (CAMS) aerosol re-analysis data (*Inness et al.*, 2013) to assess the results of the $AAOD_\mathrm{BC}$ retrieval methodology. The CAMS re-analysis assimilates satellite data into a data assimilation system and global model to correct for model departures from

observational data (*Bellouin et al.*, 2013; *Inness et al.*, 2013). The re-analysis data provides not only total AOD at 469, 550, 670, 865, and 1240 nm but also the AOD of five aerosol species: mineral dust, sea salt, sulphate, BC, and OM at 550 nm. Mineral dust and sea salt are being separated into three different size classes each, and BC and OM are distinguishable by hydrophilic and hydrophobic properties (*Bellouin et al.*, 2013).

### 3 Results

### 3.1 AERONET statistics

For this study, we have selected AERONET sites downwind of the major dust sources in Africa and Asia. We will refer to the two regions as Saharan and Asian for the remainder of this work. Details on the stations are provided in Table 2. An overview of the mean AOD and PLDR at 1020 nm as well as the FMF for the two regions are provided in the histograms in Figure 2 and in Table 2. While both regions show comparably similar features in the histograms of AOD (with larger mean

values for Saharan stations), there is a clear difference in the distribution and mean values of PLDRs: Saharan stations most



of the time show values above 0.25 while values below 0.15 form the majority of observations at Asian stations. The latter also show a considerable number of cases (30%) with $\delta_{1020} < 0.02$, for which we assume that dust is completely absent. The distribution of $\delta_{1020}$ is directly related to the contribution of mineral dust at the respective sites which is also reflected in the FMF. Most observations at Saharan sites show $FMF < 0.2$ with highest values of 0.4 while the observations at Asian sites

show a broad distribution across all possible values with peaks at 0.3 and 0.5. Overall, the two regions allow for assessing the methodology proposed here in situations dominated by mineral dust (Saharan) as well as in dusty mixtures with a broad range of dust/non-dust mixing ratios.

The effect of the different dust contributions is also apparent in the histograms of extinction and absorption Ångström exponents for the two regions in Figure 3. An absorption Ångström exponent close to unity is the theoretical value for black

carbon (*Bergstrom*, 1973; *Bohren and Huffman*, 1983) while higher values of 1.5 have been associated with biomass burning and those exceeding 2.0 represent an increasing contribution of mineral dust (*Bond et al.*, 2013). Due to the dominance of mineral dust, Saharan observations show a weak spectral dependence of AOD while a broad range of values between 1 and 4 is found for the absorbing Ångström exponent in Figure 3b. Similar values between 1.5 and 3.5 have been reported by *Russell et al.* (2010) for Arabian and Saharan dust. The large absorbing Ångström exponents result from the strong spectral

dependence of the absorbing properties of mineral dust (*Müller et al.*, 2009; *Petzold et al.*, 2009). This effect is also reflected in the spectral variation of the single-scattering albedo (not shown). The observations at Asian sites show a higher extinction Ångström exponent peaking at 1.0 to 1.25 and a lower absorption Ångström exponent with a maximum between 1.0 and 1.5. Consequently, this leads to a less pronounced spectral dependence of the single-scattering albedo (not shown). Figure 3 confirms the first impression provided by Figure 2 regarding the different contribution of mineral dust to the total AOD in the

two regions.

The dust ratio $\chi_d$ as derived using Eq. (10) for the observations in the two regions is presented in Figure 4. The general shape of the histograms of $\chi_d$ resembles that of $\delta_{1020}$ in Figure 2b. The crucial difference is that PLDR marks a proxy of the contribution of mineral dust to the measurement of the backscatter coefficient as performed with lidar while $\chi_d$ quantifies the contribution of mineral dust to the AERONET sun/sky photometer measurement of columnar AOD. The large occurrence rate

of $\chi_d$ of zero and unity refers to observations of $\delta_{1020}$ below and above the thresholds for non-dust and dust particles, respectively. Figure 4 reveals an occurrence rate of 47% and 4% for pure dust conditions for Saharan and Asian sites, respectively, when considering cases with $\chi_d > 0.9$ as pure dust. It also shows that situations with dust contributions below 50% are rare for the Saharan stations while they are most common for the Asian sites. This suggests that the selected data set includes a wide spread of situations for testing the methodology proposed here.

A closer view on the relationship between $\delta_{1020}$ and $\chi_d$ is provided in Figure 5. The figure shows the spread of $\chi_d$ that is introduced when transforming the simple theoretical relationship of Eq. (2) for lidar backscatter measurements (*Shimizu et al.*, 2004; *Tesche et al.*, 2009b) to extinction data by means of Eq. (10). Depending on the value of the total lidar ratio for the aerosol mixture with respect to the reference value for pure dust conditions (Table 1), $\chi_d$ is either increased or decreased with respect to $R_d$. Figure 5 shows that $\chi_d$ is almost exclusively larger than $R_d$ for observations at Asian sites as the majority of

AERONET-derived values of $S$ is smaller than the reference value for Asian dust presented by *Shin et al.* (2018) (not shown).



The same is the case for the Saharan observations with $\delta_{1020} < 0.2$ while above that value, $\chi_d$ is spread evenly to both sides of $R_d$. The latter is related to the fact that the frequency distribution of $S$ for the Saharan observations peaks around the value for pure Saharan dust of 54 sr (not shown) and the generally larger occurrence rate of pure-dust cases used to define the reference value in *Shin et al.* (2018). When considering the effect of FMF (not shown), we find that low values of FMF are generally

linked to higher values of $\delta_{1020}$ for both Asian and Saharan sites. However, there are occasional cases for which low FMF can be found for low values of $\delta_{1020}$ which might introduce artifacts when using FMF as a means for separating dusty from dust-free aerosol conditions.

### 3.2   Coarse-mode AOD versus dust AOD

A comparison of the coarse-mode AOD as provided by AERONET to the dust AOD obtained using Eqs. (6) and (6), respec-
tively, is presented in Figure 6. Unsurprisingly, we find that lower coarse-mode and dust AODs are related to lower coarse-mode volume concentrations (not shown) For the Asian stations, we find that coarse-mode AOD tends to overestimate the contribution of mineral dust to AOD. The effect is particularly pronounced at AODs below 0.5 at 1020 nm and coarse-mode volume concentrations below 0.5. This means that other coarse particles, such as marine aerosols, are likely to be present under these conditions. As a consequence, fine-mode AOD, if used as proxy for non-dust aerosols, would lead to a systematic underestima-
tion of the contribution of non-dust aerosol to total AOD. For the Saharan stations, coarse-mode AOD is found to be suitable proxy for dust AOD. However, coarse-mode AOD shows few values below 0.1 while dust AOD can be as low as zero. Because the concentration of fine-mode aerosol is generally small at the selected Saharan sites, any comparison to non-dust AOD is inconclusive. In contrast to the Asian sites, the AOD related to fine-mode or non-dust particles is generally much lower to that of coarse-mode or dust particles, respectively (not shown).

We conclude that that coarse-mode AOD and dust AOD cannot necessarily be considered as synonymous. This needs to be kept in mind when using AERONET observations in the calibration/validation of spaceborne remote-sensing observations and aerosol transport modelling - particularly for locations with a high occurrence rate of complex aerosol mixtures.

### 3.3   AERONET-derived $AAOD_{BC}$ and model assessment

Figure 7 presents the connection between $AAOD$ and $AAOD_{BC}$ at the standard AERONET wavelengths for observations
at the Asian and Saharan sites. $AAOD_{BC}$ has been obtained from the non-dust $AAOD$ following Eq. (15). We found that absolute values of $AAOD$ are generally larger for Asian compared to Saharan sites and that the contribution of mineral dust to aerosol absorption at all wavelengths is generally larger at Saharan compared to Asian sites. A majority of $AAOD_{BC}$ values at Asian sites follows the theoretical curve for dust-free situations (i.e. with $\chi_{dust} = 0$) and the connection between $AAOD$ and $AAOD_{BC}$ is almost linear – particularly at longer wavelengths and larger $AAOD_{BC}$. For the same $AAOD$, a larger dust ratio
$\chi_{dust}$ leads to a smaller $AAOD_{BC}$ and a corresponding observation is located further away from the solid line (not shown). The abundance of pure dust conditions at the Saharan sites therefoe leads to the larger spread of $AAOD_{BC}$ in Figure 7, that is particularly pronounced at 440 nm.





To evaluate the quality of the methodology for retrieving $AAOD_{BC}$, we have compared AERONET-derived values to the ones provided by CAMS aerosol reanalysis data for the sites considered in this study. We have investigated cases in which total AOD from AERONET and CAMS agree within 30%, 10%, and 5% of each other. We use these thresholds as a crude measure of consistency between the two data sets and to assure that we consider cases in which the modelled aerosol situation is most likely resembling observations. The plots in Figure 8 show a very different situation for the Asian and Saharan sites: the former

show correlated results and slopes of the linear fit that are reasonably close to the 1:1 line (particularly when requiring less than 5% difference in measured and modelled $AOD$, while the latter suggest that the CAMS $AAOD_{BC}$ is strongly underestimating the contribution of BC to light absorption in mixed Saharan dust plumes. The best model resemblance of $AAOD_{BC}$ is found for Dakar, where local pollution has a much stronger effect on aerosol composition than at the other Saharan sites (*Petzold et al.*, 2011). This suggests that $AAOD_{BC}$ as derived here from AERONET observations is more likely to describe aerosol absorption

in anthropogenic pollution than in biomass-burning.

We have presented a very selective analysis of AERONET observations as a proof of concept of the proposed methodology. More conclusive findings will require a thorough investigation of observations at a much larger set of AERONET sites.

## 4 Summary and conclusions

We have presented a methodology to separate the contribution of dust and non-dust aerosol to total $AOD$ measured with

AERONET instruments based on lidar parameters provided in the version 3 level 2.0 inversion product. We showed how to derive the $AAOD$ related to the non-dust component as well as to the BC fraction. We have analysed AERONET time series at six sites that are frequently affected by Asian or Saharan dust, respectively. We found that coarse- and fine mode $AOD$ cannot always be considered as synonymous with the $AOD$ related to dust and non-dust aerosol, respectively. We note that our methodology is the first to enable such a differentiation solely on products provided by AERONET. We have compared

retrieved values of $AAOD_{BC}$ to collocated model results provided by the CAMS aerosol reanalysis. This comparison has been restricted to only those AERONET-CAMS matches, for which total AOD agrees within 30% or better. We find that our methodology for obtaining $AAOD_{BC}$ from AERONET provides values that resemble CAMS aerosol modelling for Asian sites. Little correlation was found for Saharan sites that are not frequently affected by a considerable contribution of anthropogenic pollution. This suggests that $AAOD_{BC}$ as derived here is less useful for observations of biomass-burning smoke – though the

currently investigated data set has been far too small to draw a robust conclusion.

We consider the presented methodology as a useful tool for a more detailed calibration and validation of spaceborne remote-sensing observations and aerosol dispersion modelling with AERONET measurements. It will be particularly valuable at locations with a frequent occurrence of complex mixtures of mineral dust and anthropogenic pollution, e.g. east Asia or southern Europe but also individual highly polluted big cities downwind from major deserts.

*Data availability.* The data used in this work are freely available through the AERONET portal at http://aeronet.gsfc.gov/.





*Author contributions.*  SKS, MT, and YN had the idea for this study. SKS and MT performed the data analysis and prepared the figures. All authors contributed to the discussion of the findings and the preparation of the manuscript.

*Competing interests.*  The authors declare that no competing interests are present.

5   *Acknowledgements.*  We thank the PIs of the AERONET site used in this study for maintaining their instruments and providing their data to the community. We also would like to thank AERONET for their continuous efforts in providing high-quality measurements and derivative products. All data used in this work can be accessed through the AERONET web page http://aeronet.gsfc.gov/. This work was supported by the Korean Meteorological Administration Research and Development program under Grant KMI (2018-04010).





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





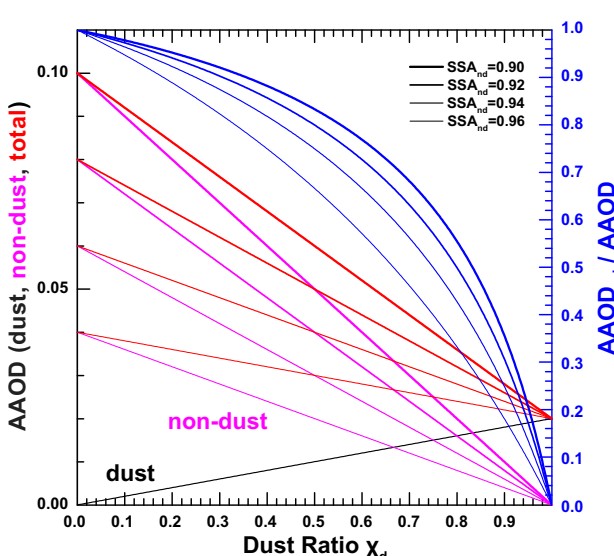

**Figure 1.** Change in $AAOD$ (red), $AAOD_\mathrm{d}$ (black), and $AAOD_\mathrm{nd}$ (magenta) with dust ratio $\chi_\mathrm{d}$ for an aerosol mixture of dust ($\omega_\mathrm{d} = 0.98$) and non-dust ($\omega_\mathrm{nd} = 0.90 - 0.96$) and an $AOD$ of unity. The blue lines mark the contribution of non-dust aerosol to $AAOD$ for different values of $\omega_\mathrm{nd}$.

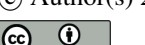



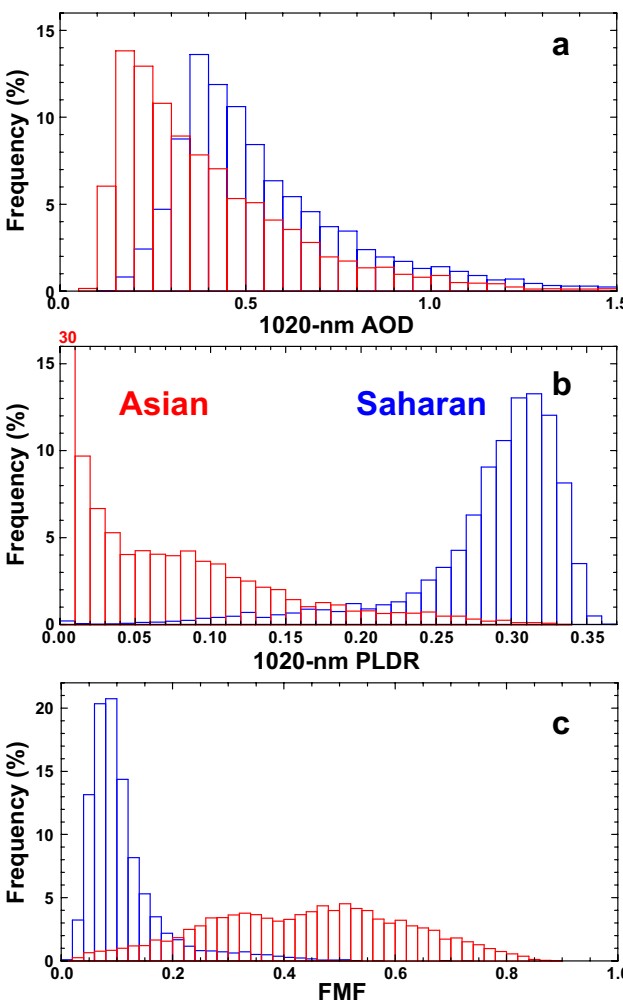

**Figure 2.** Histograms of the 1020-nm $AOD$ (a), 1020-nm PLDR (b), and $FMF$ (c) for the considered AERONET stations affected by Saharan (blue) and Asian dust (red). Coloured numbers provide the values for bars that exceed the scale. Details on the considered AERONET stations and mean values are provided in Table 2.




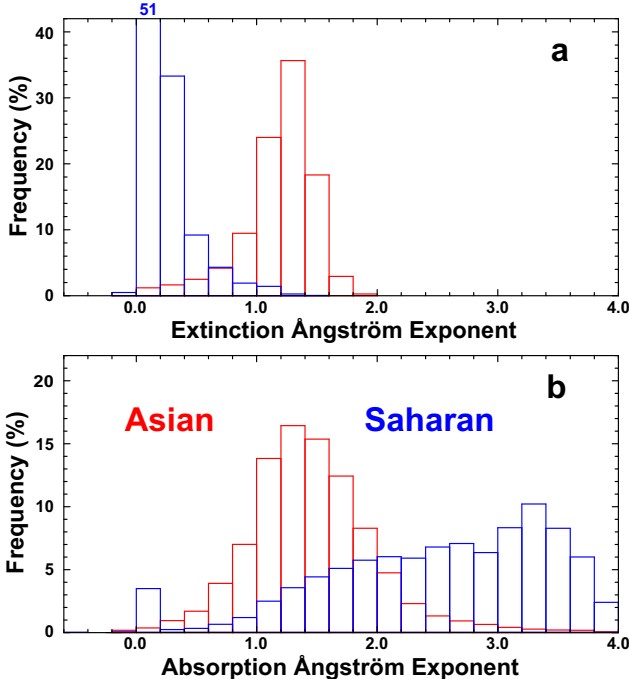

**Figure 3.** Histograms of the 440-870-nm extinction (a) and absorption (b) Ångström exponents for the considered AERONET stations affected by Saharan (blue) and Asian dust (red). Coloured numbers provide the values for bars that exceed the scale. Details on the considered AERONET stations in Table 2.

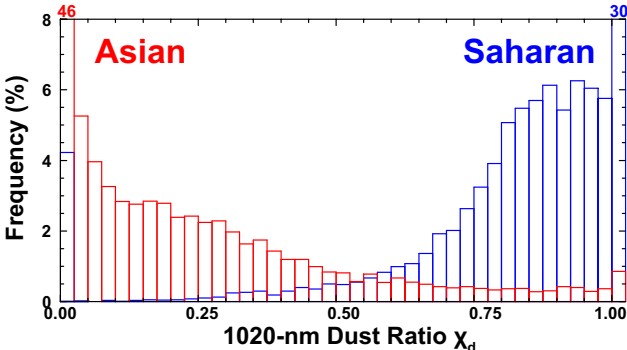

**Figure 4.** Histograms of $\chi_d$ for the considered AERONET stations affected by Saharan (blue) and Asian dust (red). Coloured numbers provide the values for bars that exceed the scale.





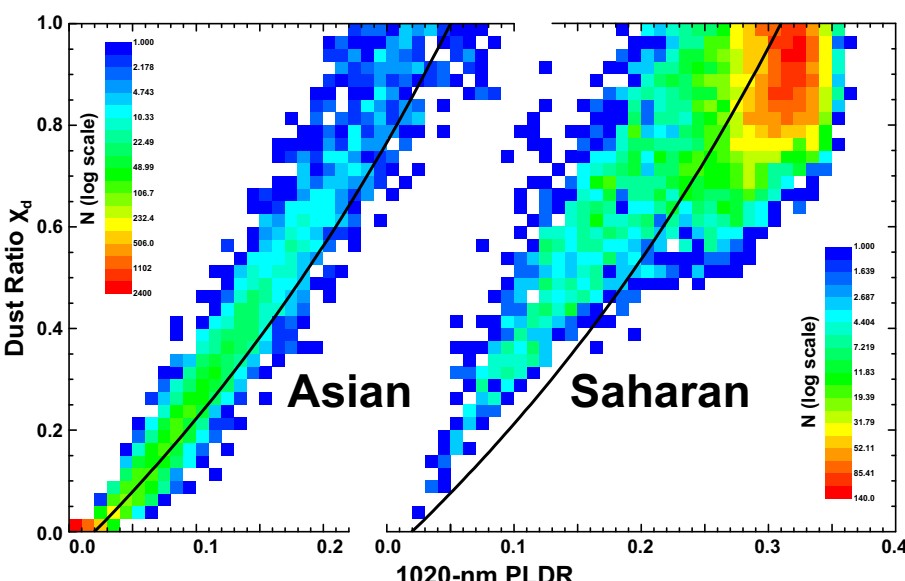

**Figure 5.** 2d histograms of 1020-nm PLDR and $\chi_d$ for the considered AERONET stations affected by Asian (left) and Saharan dust (right). The black lines refer to the backscatter-related dust ratio $R_d$.



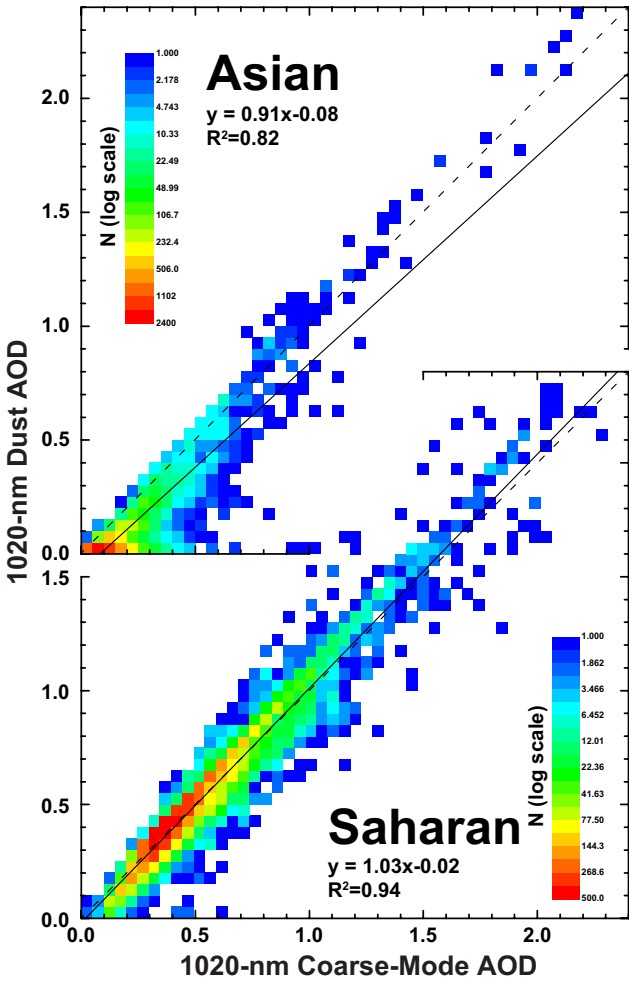

**Figure 6.** 2d histograms of 1020-nm coarse-mode $AOD$ and dust-related 1020-nm $AOD$ for the considered AERONET stations affected by Asian (upper panel) and Saharan dust (lower panel).





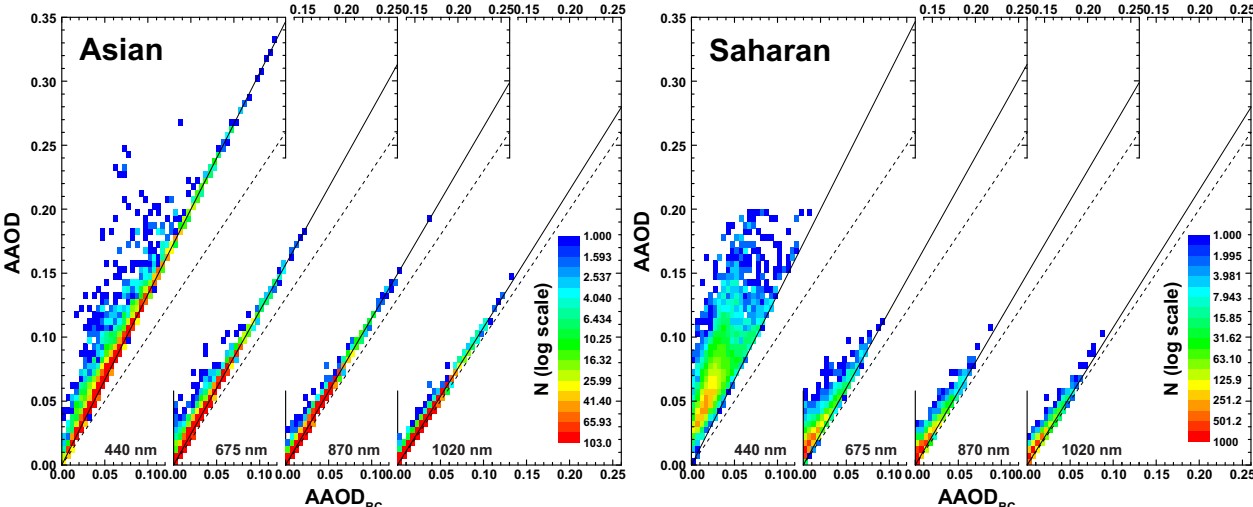

**Figure 7.** 2d histograms of $AAOD$ and $AAOD_{BC}$ at the four AERONET standard wavelengths for the Asian (upper panel) and Saharan (lower panel) stations considered in this study. Solid lines refer to the theoretical values of $AAOD_{BC}$ (using Eq. (15) and the values in Table 1) in the absence of mineral dust. Dashed lines mark the 1:1 line. Observations would follow this slope only if BC was a perfect absorber, i.e. if all absorption in the non-dust fraction would be due to BC or $\omega_{BC} = 0$.





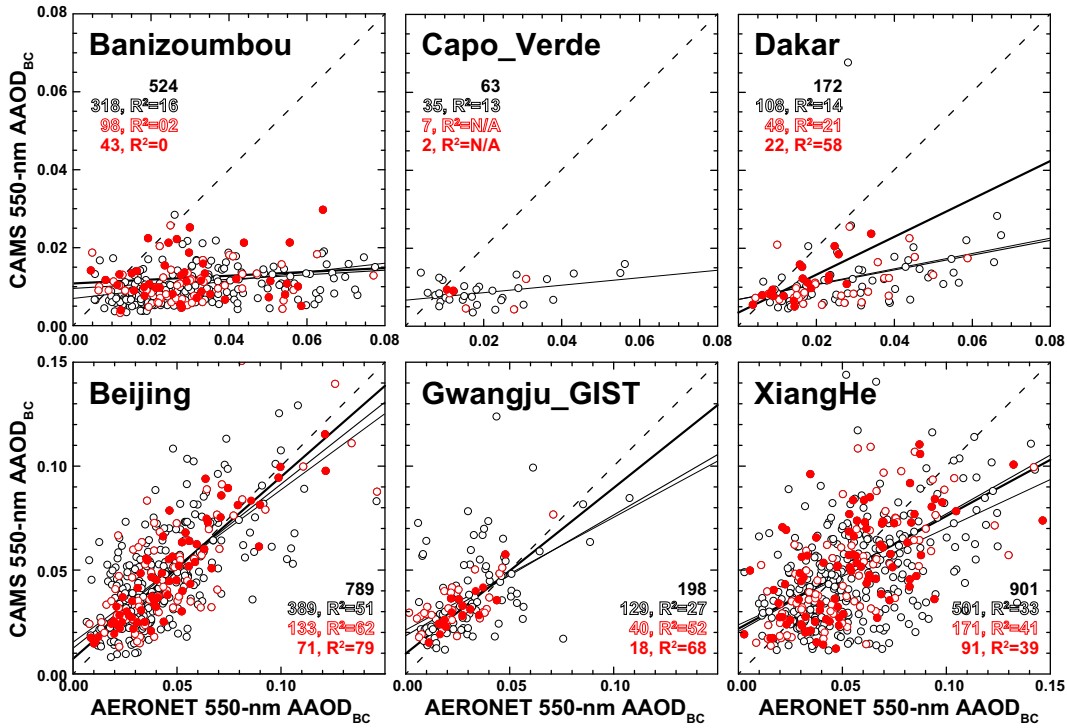

**Figure 8.** Comparison of AERONET-derived $AAOD_{BC}$ with CAMS model estimates for the sites listed in Table 2 for cases in which the total $AOD$ from CAMS and AERONET agrees within 30% (black circles, thin lines), 10% (red circles, medium lines), and 5% (red dots, bold lines). Numbers in the plots refer to the total number of collocated points and the number of matches with the given AOD agreement. Dashed lines mark the 1:1 line. Solid lines are linear fits of the data. Numbers in the plots refer to the number of collocations and squared correlation coefficients for all cases (solid black, no $R^2$ given) and those with an AOD agreement within 30% (open black), 10% (open red), and 5% (solid red).



**Table 1.** List of input parameters used for the retrieval of $AAOD_{\mathrm{nd}}$ and $AAOD_{\mathrm{BC}}$ in this study. Values of $\delta$ at 1020 nm are used for the separation of optical properties of dust and non-dust particles. The dust-related Ångström exponent is needed to transform findings at 1020 nm to other wavelengths. The values of $\omega_{\mathrm{d}}$ and $\omega_{\mathrm{BC}}$ are used to retrieve $AAOD_{\mathrm{nd}}$ and $AAOD_{\mathrm{BC}}$, respectively.

| Paramater | Symbol | Value | | | | Reference |
|---|---|---|---|---|---|---|
| | | 440 nm | 675 nm | 870 nm | 1020 nm | |
| total AOD | $AOD$ | | | | | |
| total PLDR | $\delta$ | | from individual AERONET | | | |
| total lidar ratio | $S$ | | version 3 level 2.0 measurements | | | |
| total SSA | $\omega$ | | | | | |
| non-dust PLDR | $\delta_{\mathrm{nd}}$ | - | - | - | $0.02 \pm 0.01$ | *Shimizu et al.* (2004) |
| dust PLDR (Asian) | $\delta_{\mathrm{d}}$ | - | - | - | $0.30 \pm 0.04$ | *Shin et al.* (2018) |
| dust PLDR (Saharan) | $\delta_{\mathrm{d}}$ | - | - | - | $0.31 \pm 0.03$ | *Shin et al.* (2018) |
| dust lidar ratio (Asian) | $S_{\mathrm{d}}$ | - | - | - | $44 \pm 6$ sr | *Shin et al.* (2018) |
| dust lidar ratio (Saharan) | $S_{\mathrm{d}}$ | - | - | - | $54 \pm 9$ sr | *Shin et al.* (2018) |
| dust Ångström exponent | $å_{\mathrm{d}}$ | | $0.06 \pm 0.21$ | | | *Tesche et al.* (2009a) |
| dust SSA | $\omega_{\mathrm{d}}$ | 0.94 | 0.98 | 0.99 | 0.99 | *Eck et al.* (2005); *Yu et al.* (2006) |
| BC SSA | $\omega_{\mathrm{BC}}$ | $0.25 \pm 0.13$ | $0.17 \pm 0.01$ | $0.13 \pm 0.03$ | $0.07 \pm 0.02$ | *Haywood and Ramaswamy* (1998) |

**Table 2.** Overview of the AERONET sites included in this study in terms of location, length of time series and number of available version 3 level 2.0 data points. The last three columns refer to mean values and standard deviation of $AOD_{1020}$, $\delta_{1020}$, and $FMF$ for the respective sites and regions. The figures in this work refer to the combined Asian and Saharan data sets.

| Station | Location | Period | $N$ | $AOD_{1020}$ | $\delta_{1020}$ | $FMF$ |
|---|---|---|---|---|---|---|
| Beijing | 39.98 °N, 116.38 °E | 2001–2018 | 2713 | $0.45 \pm 0.29$ | $0.06 \pm 0.07$ | $0.42 \pm 0.17$ |
| Gwangju_GIST | 35.23 °N, 126.84 °E | 2004–2018 | 956 | $0.25 \pm 0.12$ | $0.06 \pm 0.07$ | $0.51 \pm 0.19$ |
| XiangHe | 39.75 °N, 116.96 °E | 2001–2018 | 4300 | $0.41 \pm 0.25$ | $0.06 \pm 0.07$ | $0.44 \pm 0.18$ |
| **combined Asian** | | 2001–2018 | 7969 | $0.41 \pm 0.26$ | $0.06 \pm 0.07$ | $0.44 \pm 0.18$ |
| Banizoumbou | 13.55 °N, 2.67 °E | 1995–2018 | 4217 | $0.60 \pm 0.31$ | $0.29 \pm 0.05$ | $0.11 \pm 0.08$ |
| Capo_Verde | 16.73 °N, 22.94 °W | 1994–2018 | 1689 | $0.55 \pm 0.25$ | $0.30 \pm 0.05$ | $0.09 \pm 0.04$ |
| Dakar | 14.39 °N, 16.96 °W | 1996–2018 | 4118 | $0.54 \pm 0.28$ | $0.28 \pm 0.06$ | $0.12 \pm 0.08$ |
| **combined Saharan** | | 1994–2018 | 10024 | $0.57 \pm 0.29$ | $0.29 \pm 0.05$ | $0.11 \pm 0.07$ |