# Peer review of "Technical note: Absorption aerosol optical depth components from AERONET observations of mixed dust plumes"

_Atmospheric Measurement Techniques, 2018_

## Referee Comment (RC1) · Anonymous Referee #3 · 6 Nov 2018

Review on

Technical note: Absorption aerosol optical depth components from AERONET observations of mixed dust plumes

By Shin et al.

AMT-2018-311

General comments

This article addresses a problem of aerosol absorption (AAOD) separation in the mixed aerosol plumes, which is of high interest of scientific community. Authors suggest

using previously developed lidar based technique but applied to standalone AERONET observations, to separate aerosol absorption between dust and Black Carbon particles. The methodology description and evaluation for Black Carbon AAOD separation are rather brief. I would not recommend this note for publication in its actual state; it needs major revisions, since authors in my opinion had missed some crucial points in their study.

There are two major issues:

1. Dust/non-dust properties separation

Authors use a lidar based method to estimate the proportion of the desert dust in the mixture using only AERONET data, and yet do not show any comparison to an AERONET provided so called "percentage of spherical particles", which by definition gives the proportion between spherical (i.e. non-dust) and non-spherical (dust) particles (see Dubovik et al., 2006). To my strongest belief, any new methodology should be compared to an existing one, in order to estimate its scientific value. At this point it is absolutely unclear, which advantage new method proposes, in case when lidar is not available, in comparison with already existing AERONET provided product.

2. Black Carbon properties separation

The major part of the method description operates in terms of dust and non-dust particles. To me, method to derive a black carbon content from a non-dust AOD, taking black carbon SSA as a coefficient describing the amount of BC in the mixture has no sufficient support. Formula 15 in a given form doesn't have much physical sense and referred papers do not contain any similar equations. Values selected for black carbon SSA are more suitable for laboratory measurements they were taken from. Such low values could be observed only in the immediate vicinity to the particle origin, which is not the case in a situation with aerosol transport. The fact that proposed method significantly overestimates BC over Saharan desert sites proves that such assumption could be made only for selected sites or cases. I would recommend changing SSA value

over Saharan sites to typical for Biomass Burning aerosol to see if correlation could be improved. This would help to support some of the conclusions that were made.

Specific comments:

Page 3. Lines 7–8. "The AERONET inversion is performed for measurements with a 440-nm AOD larger than 0.4" Phrase is not actually correct, AERONET inversions are performed at any AODs, yet it is true that quality assured SSA retrievals (level 2) have a threshold on minimum AOD values.

Page 5. Line 1. "we use the Angstroom exponent ad = 0.06±0.21", it is not clear which exact value or values were used. Were they selected within the given range? Was the same value used for all the cases or it was varied? There is a significant difference in the formula 8 behaviour having angstrom -0.15 or 0.27.

Page 8. Line 19. "We conclude that that coarse-mode AOD and dust AOD cannot necessarily be considered as synonymous." I would really like to see same comparison made with non-spherical part of coarse AOD, which can easily be obtained by using AERONET provided percentage of spherical particles. Such comparison would be more correct.

---

## Referee Comment (RC2) · Anonymous Referee #4 · 8 Nov 2018

"General comments" The analysis of the AAOD components of mixed dust plumes is an important topic for scientific community and the proposed method to apply a lidar technique to the AERONET v3 inverse products could be very interesting. However, the authors should clarify key issues to make robust and rigorous the approach presented in this note. Besides, major revision of the overall presentation should be properly addressed before the publication.

"Specific comments": 1) The level 2.0 assures the quality level of of direct and inverse AERONET products. What about the level (1.0, 1.5 or 2.0) of the AERONET data used in this study? 2) Have the authors used the AERONET's recommended loading con-

straint (AOD>0.4 at 440nm) for inverse products? 3) Shin et al., 2018 reported specific conditions for PLDR in case of pure mineral dust: "To select observations representative of pure mineral dust conditions, only AERONET data with a 440/870 nm Ångström exponent below 0.4 and a fine-mode fraction below 0.10 have been selected in this study." The authors should explain how they overcome both these conditions considering the values of PLDR reported in Shin et al., 2018 in different FMF and Ångström exponent domains. 4) The methodology for the retrieval of AAOD of dust and BC components is based on Equation (6). From the description of methodology, the authors assumed the same single aerosol layer of depth h for both aerosol mixing and dust. Furthermore, the integrated-values of the extinction coefficient for the mixed plume and the dust component were solved by assuming the extinction coefficient constant in the layer. The AOD AERONET product represents the integration of the vertically varying extinction coefficient in the entire atmospheric column. The authors should explain how the integration domain of the total columnar mixing aerosol (AOD) can be limited to a single layer in which the aerosol mixing and the dust component are limited. Furthermore, the assumption of vertically non-varying extinction coefficient should be in-depth explained.

"Technical corrections" p. 3 l. 4 Please, explicit the amount of AERONET bands (not 'several') p. 4 l. 11 Probably, a bracket is missing p. 8 l. 9 'Eqs (6) and (6)'. Please, control the reference to equations. p. 8 l. 31 'therefoe'
* * *

---

## Referee Comment (RC3) · Anonymous Referee #1 · 14 Nov 2018

Manuscript touches important problem: separation the components of aerosol mixture characterized by different absorbing properties basing on AERONET measurements. Manuscript is clearly written, provides new useful results and is suitable for publishing in AMT.

Referees #3 and #4 provided extended comments, so I have just several notes.

For separation of dust and non dust components authors use depolarization ratio recalculated from AERONET inversions. So I wonder if such separation can be done directly from spheroids volume fraction. I think authors should explain necessity of using lidar related characteristics.

[Figure]

p.2 ln.10. "BC as emitted from incomplete anthropogenic combustion or biomass burning is generally considered the main light absorber among atmospheric aerosols" Actually contribution of brown carbon to absorption can be also significant. I think authors should comment it.

p.4 ln.12. Authors assume depolarization of non dust particles to be of 0.02. Actually depolarization ratio of smoke varies in a wide range and can exceed 10% (Butron et al, 2012). How such variations may influence results?

Eq(5). Authors introduce the layer thickness "h" but looks like never use it later.

p.4.ln.28. "We take the values of 44 sr and 54 sr for Asian and Saharan dust, respectively" Lidar ratios even for pure dust can vary in a wide range. This should be commented.

---

## Short Comment (SC1) · 15 Nov 2018

Could authors give more explanations or a reference to "the mixing rule" that is used to derive Eq. (12) in the line # 15?

Sorry, Eq. (15) in the line #25 is not obvious as well.
* * *

---

## Referee Comment (RC4) · A. Lyapustin (Referee) · 16 Nov 2018

P2, Ln15: "Russell et al. (2010) utilized AERONET-retrieved SSA, AAOD, and absorption Ångström exponent (AAE) as indicator to separate the contributions of BC, organic matter (OM), and mineral dust to the absorbing aerosol fraction." That is incorrect: This work suggested a preferred classification scheme, not a "separation".

P2, L21: ". . . and to infer the fine or coarse mode fraction in the aerosol size distribution" What does it mean?

P3, L10: "The uncertainty in SSA is expected to be of the order of 0.03 (Holben et al.,

1998)." The reference is incorrect.

Eq. (4) in my view is incorrect. The correct equation for the measured backscattering should be something like Beta = f_dust*Beta_dust + f_nd*Beta_nd, where f_dust=AOD_dust/AOD, AOD=AOD_dust+AOD_nd. If phase functions of dust and non-dust are very different, and they usually are because of the particle size difference, then Eq. (4) is not correct.

I may be wrong as I did not work with the lidars before. But if I am correct, then this error (or assumption) propagates in all derivations below. I suggest a major revision with authors addressing this key point first.

Alexei.

———————————————————

---

## Referee Comment (RC5) · Anonymous Referee #5 · 16 Nov 2018

"Absorption aerosol optical depth components from AERONET observations of mixed dust plumes" by Sung-Kyun Shin, Matthias Tesche, Detlef Müller, and Youngmin Noh

Reviewer's comments

Manuscript presents a new technique to distinguish between contributions of dust and non-dust aerosols to the total aerosol optical depth (AOD) measured by AERONET. The approach is based on utilizing the particle linear depolarization ratio (PLDR) which is now available as one of aerosol products in AERONET Version3. Due to PLDR sensitivity to aerosol particle shape, the dust/non-dust separation is based on the particles shape differences rather than on difference in aerosol size. This, for example, allows

excluding contribution of coarse spherical particles from dust AOD component.

I believe that the subject of manuscript is in scope of AMT, it is very well written, and presented results are interesting. I recommend it to be published with minor revisions.

Comments.

1. My main comment is regarding Eq. (15). It is not clear how it was derived and what model of non-dust aerosol component was assumed. Equation suggests that black carbon (BC) is not the only absorber in non-dust component (both BC and non-dust single scattering albedos are present in (15)). However if the second absorber is present its properties need to be described. In addition, the right side of the equation suggests that AAODnd should be equivalent to BC extinction AOD, which is not obvious without knowing what model of non-dust component was employed. I was able to derive the equation assuming the presence of the second very low absorbing component in non-dust component mixture. However, it still not clear how it was done by authors. I suggest to include discussion on how the Eq. (15) was derived and what assumptions were used.

2. Page 3, between lines 5 and 10 authors write: "The AERONET inversion is performed for measurements with a 440-nm AOD larger than 0.4". Actually inversion is performed for all the values of AOD, but only inversions for AOD (440)>0.4 are included in Level 2 product.

3. Page 3, line 10. For reference on uncertainty in SSA retrievals is more appropriate to use Dubovik (2000) uncertainty paper.

4. Page 5. "Re-arranging Eq. (11)". I think it should be Eq. (12).

5. Page 8. "using Eqs. (6) and (6),". Probably typo.

6. Page 8. Why analysis of AAOD (BC) was conducted at all four AERONET wavelength? The earlier discussion suggested using just 1020 nm PLDR product as being more reliable.

7. Figure 7 caption. Upper and lower panels should be probable left and right panels.

8. You may consider using AERONET SDA product to compare to dust AOD in addition to coarse mode AOD inferred from particle size distribution retrievals.
* * *

---

## Referee Comment (RC6) · Anonymous Referee #6 · 21 Nov 2018

Review on Technical note: Absorption aerosol optical depth components from AERONET observations of mixed dust plumes By Shin et al. AMT-2018-311

The authors suggest a methodology to separate the contributions of dust and non-dust aerosol to total AOD measured with AERONET sun photometers based on lidar parameters. I think that the methodology can be a useful tool for interpreting the data of ground-based photometric observations and validating the spaceborne remote sensing observations. The manuscript is within the scope of AMT.

General comments:

The approach, suggested by the authors, is designed to retrieve AAOD for non-dust

aerosol in mixed dust plumes. The formulas, they present, pertain to the case when aerosol is considered as a vertically homogeneous layer. Even if the authors will take into consideration the comments of other reviewers regarding these formulas, the approach will still rely upon the assumption on vertical homogeneity. Evidently, in passing to measurement results, we cannot expect that this assumption will be fulfilled: dust and non-dust aerosols may be separated in altitude. Therefore, if this methodology is to be considered as a "guide for action", a cycle of additional numerical and field experiments is at least required. Of course, that is unnecessary to be done within the given manuscript, but does need to be done in the future. In addition to the issue of vertical inhomogeneity, it would be interesting to see the results concerning the following aspects: (1) absorption in not only black carbon, but also in brown carbon, which is shifted to UV region of the spectrum; (2) effect of uncertainty in retrieving the aerosol characteristics by the standard AERONET algorithm on the AAOD retrievals, based on this method

All technical shortcomings, noted by me, have already been indicated by other reviewers and are hoped to be accounted for in the final version of the manuscript.

Please also note the supplement to this comment:
https://www.atmos-meas-tech-discuss.net/amt-2018-311/amt-2018-311-RC6-supplement.pdf

---

## Author Comment (AC1) · 7 Jan 2019

Interactive comments on "Technical note: Absorption aerosol optical depth components from AERONET observations of mixed dust plumes" by Sung-Kyun Shin et al.

Referee comments are noted in black. Our replies are given in blue.

We would like to thank all Referees for their constructive comments. Please find our point-by-point replies below. We have also attached a revised version of the manuscript with all changes marked.

**Anonymous Referee #3**

General comments

This article addresses a problem of aerosol absorption (AAOD) separation in the mixed aerosol plumes, which is of high interest of scientific community. Authors suggest using previously developed lidar based technique but applied to standalone AERONET observations, to separate aerosol absorption between dust and Black Carbon particles. The methodology description and evaluation for Black Carbon AAOD separation are rather brief. I would not recommend this note for publication in its actual state; it needs major revisions, since authors in my opinion had missed some crucial points in their study.

There are two major issues:

1. Dust/non-dust properties separation

Authors use a lidar based method to estimate the proportion of the desert dust in the mixture using only AERONET data, and yet do not show any comparison to an AERONET provided so called "percentage of spherical particles", which by definition gives the proportion between spherical (i.e. non-dust) and non-spherical (dust) particles (see Dubovik et al., 2006). To my strongest belief, any new methodology should be compared to an existing one, in order to estimate its scientific value. At this point it is absolutely unclear, which advantage new method proposes, in case when lidar is not available, in comparison with already existing AERONET provided product.

We agree with the reviewer that new methods have to be compared with existing ones. However, we think that the suggested comparison of percentage of spherical particles (sphericity) to the contributions of dust/non-dust as derived using the methodology presented in our paper is not useful and would rather add confusion than clarification. Two points support our reasoning:

1.  Sphericity is a retrieval parameter of the AERONET inversion rather than a physically meaningful quantity. As such, it is specific to the AERONET retrieval. In contrast to PLDR, it cannot be compared to independent measurements.

2.  Sphericity is no longer included as an output parameter of the AERONET version 3 retrieval. For the data points considered in our study, we have collated version 3 PLDRs with version 2 sphericity to investigate a possible connection between the two parameters. Figure 1 shows that while low values of sphericity are generally connected to higher PLDRs, there is not clear relationship of decreasing PLDRs with increasing sphericity. In contrast, there is a clear link between PLDR and the contribution of non-spherical particles as documented in the literature.

The advantage of the new method is that it relies on physically meaningful parameters that can be observed through independent measurements. It is based solely on AERONET products, and thus, does not require co-located lidar measurements.

[Figure]

**Figure 1:** Correlation between sphericity (from AERONET version 2) and 1020-nm PLDR (from AERONET version 3) for level 2.0 data from Beijing, Cape Verde, and Dakar.

We have added the text below to Section 2.1. to refer to an earlier study that investigated the consistency between PLDRs as retrieved from AERONET and lidar observations:

*"Noh et al. (2017) investigated the reliability of the PLDR retrieved from AERONET sun/sky radiometer observations and found the strongest correlation between the 1020-nm PLDR inferred from AERONET data and the 532-nm PLDR from lidar observations."*

Noh, Y., Müller, D., Lee, K., Kim, K., Lee, K., Shimizu, A., Sano, I., and Park, C. B.: Depolarization ratios retrieved by AERONET sun–sky radiometer data and comparison to depolarization ratios measured with lidar, Atmos. Chem. Phys., 17, 6271-6290, https://doi.org/10.5194/acp-17-6271-2017, 2017.

**2. Black Carbon properties separation**

The major part of the method description operates in terms of dust and non-dust particles. To me, method to derive a black carbon content from a non-dust AOD, taking black carbon SSA as a coefficient describing the amount of BC in the mixture has no sufficient support. Formula 15 in a given form doesn't have much physical sense and referred papers do not contain any similar equations. Values selected for black carbon SSA are more suitable for laboratory measurements they were taken from. Such low values could be observed only in the immediate vicinity to the particle origin, which is not the case in a situation with aerosol transport. The fact that proposed method significantly overestimates BC over Saharan desert sites proves that such assumption could be made only for selected sites or cases. I would recommend changing SSA value over Saharan sites to typical for Biomass Burning aerosol to see if correlation could be improved. This would help to support some of the conclusions that were made.

We thank the reviewer for the comment. We are aware that non-dust light absorption can be related to BC, Brown Carbon (BrC) or other compounds. However, the intention of our study is to outline the general approach to retrieve non-dust AAOD from AERONET products and to present initial findings of using this method. For this, we assume that BC is the major absorber for the considered sites. This might not always be true but it is a reasonable first guess. The BC SSA we use in our work has indeed been obtained from laboratory studies. The aim of our approach is to obtain the AAOD related to BC as defined by aerosol chemistry. If we were to use values typical for biomass-burning aerosol we would run into two problems: (i) these values are defined optically and not chemically and (ii) we wouldn't get an AAOD related to BC but to an absorbing aerosol of unknown chemical composition. Consequently, a comparison of such results to modelled BC-related AAOD would be comparing apples and oranges. However, Figure 8 already indicates that the quality of the

comparison of modelled and AERONET-derived $AAOD_{BC}$ depends to a strong degree on the agreement between modelled and observed total AOD.

To account for the referee's comment and that of other referees, we have revised our statement related to BC to:

*"In dust-free conditions, BC as emitted from incomplete combustion involved in anthropogenic activities or biomass burning is generally considered the main light absorber among atmospheric aerosols (Bond and Bergstrom, 2006; Bond et al., 2013; Russell et al., 2010), and thus, the main contributor to non-dust AAOD. **The term BC refers to carbon particles with the morphological and chemical properties typical of soot particles from combustion including a black, blackish or brown substance formed by combustion (Andreae and Gelencsér, 2006). We point out that the contribution of brown carbon (BrC) to aerosol absorption can also be significant. However, we opt for a single absorbing aerosol component as it allows us to present the general idea of our new methodology in a straightforward manner."***

Andreae, M. O. and Gelencsér, A.: Black carbon or brown carbon? The nature of light-absorbing carbonaceous aerosols, Atmos. Chem. Phys., 6, 3131-3148, https://doi.org/10.5194/acp-6-3131-2006, 2006.

Specific comments:

Page 3. Lines 7–8. "The AERONET inversion is performed for measurements with a 440-nm AOD larger than 0.4" Phrase is not actually correct, AERONET inversions are performed at any AODs, yet it is true that quality assured SSA retrievals (level 2) have a threshold on minimum AOD values.

Thank you for the clarification. We have changed the statement to:

*"The level 2 product available from the AERONET portal includes inversion results for measurements with a 440-nm AOD larger than 0.4 (Dubovik et al., 2006). The AERONET inversion uses…"*

Page 5. Line 1. "we use the Angstroom exponent ad = 0.06+/-0.21", it is not clear which exact value or values were used. Were they selected within the given range? Was the same value used for all the cases or it was varied? There is a significant difference in the formula 8 behaviour having angstrom -0.15 or 0.27.

We have used the mean value without any variation. We have clarified the statement to:

*"…we use the Angstroom exponent ad = 0.06"*

Page 8. Line 19. "We conclude that that coarse-mode AOD and dust AOD cannot necessarily be considered as synonymous." I would really like to see same comparison made with non-spherical part of coarse AOD, which can easily be obtained by using AERONET provided percentage of spherical particles. Such comparison would be more correct.

Please see our reply to major issue #1.

[revised manuscript text omitted]

Re-arranging Eq. (12) gives the SSA related to non-dust particles

$$\omega_{\text{nd},\lambda} = \frac{\omega_{\lambda} - \chi_{\text{d},\lambda}\omega_{\text{d},\lambda}}{\chi_{\text{nd},\lambda}}.$$ (13)

The spectral SSA for pure dust particles is taken from the literature (see Table 1). The non-dust fraction to AAOD can now be derived as

5  $$AAOD_{\text{nd},\lambda} = (1 - \omega_{\text{nd},\lambda})AOD_{\text{nd},\lambda}.$$ (14)

We can assume that the light-absorbing features of the non-dust part of the aerosol plume are caused primarily by BC. It has been shown that BC is not an ideal light absorber, i.e., $\omega_{\text{BC},\lambda} \neq 0$, (*Bond and Bergstrom*, 2006; *Bond et al.*, 2013). **Thus**, we need to account for the SSA of BC to obtain the BC-related AAOD as:

$$AAOD_{\text{BC},\lambda} = AOD_{\text{nd},\lambda}(1 - \omega_{\text{nd},\lambda})(1 - \omega_{\text{BC},\lambda}) = AAOD_{\text{nd},\lambda}(1 - \omega_{\text{BC},\lambda}).$$ (15)

10  *Bond and Bergstrom* (2006) report on single-scattering albedos of 0.10 to 0.28 for fresh BC. Similar values for fresh BC have also been reported by *Khalizov et al.* (2009) and *Cross et al.* (2010). Here, we use values of $\omega_{\text{BC},\lambda}$ from *Haywood and Ramaswamy* (1998). The**se values** are provided together with the other input parameters in Table 1.

**2.3  Connection between $AAOD$, $AAOD_{\text{nd}}$ and $AAOD_{\text{BC}}$**

Substituting Eq. (12) in Eq. (1) leads to the equation for the AAOD (of dusty mixtures) that accounts for the contribution of
15  the different components as:

$$AAOD = (1 - (\chi_{\text{d},\lambda}\omega_{\text{d},\lambda} + \chi_{\text{nd},\lambda}\omega_{\text{nd},\lambda}))AOD.$$ (16)

[revised manuscript text omitted]

---

## Author Comment (AC3) · 7 Jan 2019

Interactive comments on "Technical note: Absorption aerosol optical depth components from AERONET observations of mixed dust plumes" by Sung-Kyun Shin et al.

Referee comments are noted in black. Our replies are given in blue.

We would like to thank all Referees for their constructive comments. Please find our point-by-point replies below. We have also attached a revised version of the manuscript with all changes marked.

**Anonymous Referee #1**

Manuscript touches important problem: separation the components of aerosol mixture characterized by different absorbing properties basing on AERONET measurements. Manuscript is clearly written, provides new useful results and is suitable for publishing in AMT.

Referees #3 and #4 provided extended comments, so I have just several notes.

For separation of dust and non-dust components authors use depolarization ratio recalculated from AERONET inversions. So I wonder if such separation can be done directly from spheroids volume fraction. I think authors should explain necessity of using lidar related characteristics.

We agree with the reviewer that new methods have to be compared with existing ones. However, we think that the suggested comparison of percentage of spherical particles (sphericity) to the contributions of dust/non-dust as derived using the methodology presented in our paper is not useful and would rather add confusion than clarification. Two points support our reasoning:

1. Sphericity is a retrieval parameter of the AERONET inversion rather than a physically meaningful quantity. As such, it is specific to the AERONET retrieval. In contrast to PLDR, it cannot be compared to independent measurements.

2. Sphericity is no longer included as an output parameter of the AERONET version 3 retrieval. For the data points considered in our study, we have collated version 3 PLDRs with version 2 sphericity to investigate a possible connection between the two parameters. Figure 1 shows that while low values of sphericity are generally connected to higher PLDRs, there is not clear relationship of decreasing PLDRs with increasing sphericity. In contrast, there is a clear link between PLDR and the contribution of non-spherical particles as documented in the literature.

The advantage of the new method is that it relies on physically meaningful parameters that can be observed through independent measurements. It is based solely on AERONET products, and thus, does not require co-located lidar measurements.

[Figure]

**Figure 1:** Correlation between sphericity (from AERONET version 2) and 1020-nm PLDR (from AERONET version 3) for level 2.0 data from Beijing, Cape Verde, and Dakar.

p.2 ln.10. "BC as emitted from incomplete anthropogenic combustion or biomass burning is generally considered the main light absorber among atmospheric aerosols" Actually contribution of brown carbon to absorption can be also significant. I think authors should comment it.

In this study we use the terminology BC for carbon particles with the morphological and chemical properties typical of soot particles from combustion including a black, blackish or brown substance formed by combustion. In this sense we determined BC as more likely to be a primary source (without mixing). We acknowledge the potential contribution of brown carbon to aerosol absorption. However, we believe that the use of one absorbing aerosol type is sufficient to present the general idea of our methodology. We have revised our statement to:

*"In dust-free conditions, BC as emitted from incomplete combustion involved in anthropogenic activities or biomass burning is generally considered the main light absorber among atmospheric aerosols (Bond and Bergstrom, 2006; Bond et al., 2013; Russell et al., 2010), and thus, the main contributor to non-dust AAOD.* ***The term BC refers to carbon particles with the morphological and chemical properties typical of soot particles from combustion including a black, blackish or brown substance formed by combustion (Andreae and Gelencsér, 2006). We point out that the contribution of brown carbon (BrC) to aerosol absorption can also be significant. However, we opt for a single absorbing aerosol component as it allows us to present the general idea of our new methodology in a straightforward manner."***

Andreae, M. O. and Gelencsér, A.: Black carbon or brown carbon? The nature of light-absorbing carbonaceous aerosols, Atmos. Chem. Phys., 6, 3131-3148, https://doi.org/10.5194/acp-6-3131-2006, 2006.

p.4 ln.12. Authors assume depolarization of non-dust particles to be of 0.02. Actually depolarization ratio of smoke varies in a wide range and can exceed 10% (Butron et al, 2012). How such variations may influence results?

We thank the authors for this comment. We are aware of detection of PLDRs that are larger than 0.02. To obtain the value used in our study, we have repeated the investigation of *Shin et al.* (2018) for AERONET stations dominated by biomass-burning smoke. For clarification, we have added the following text:

*"The latter value has been obtained from the analysis of δ derived at AERONET stations dominated by biomass-burning aerosols, analogous to the dust-focused study of* Shin et al. (2018)*."*

Eq(5). Authors introduce the layer thickness "h" but looks like never use it later.

Correct. We have introduced this parameter for the sole purpose of resolving the connection between AOD (the columnar parameter provided by sun photometer) and extinction coefficient (the height-resolved parameter provided by aerosol lidar).

p.4.ln.28. "We take the values of 44 sr and 54 sr for Asian and Saharan dust, respectively" Lidar ratios even for pure dust can vary in a wide range. This should be commented.

We thank the Referee for this comment. We are referring to the study of *Shin et al.* (2018) in which statistics of the lidar ratios at the different dust sources are presented. Nevertheless, we have revised our statement to:

[revised manuscript text omitted]

Re-arranging Eq. (12) gives the SSA related to non-dust particles

$$\omega_{\text{nd},\lambda} = \frac{\omega_\lambda - \chi_{\text{d},\lambda}\omega_{\text{d},\lambda}}{\chi_{\text{nd},\lambda}} \,. \tag{13}$$

The spectral SSA for pure dust particles is taken from the literature (see Table 1). The non-dust fraction to AAOD can now be derived as

5  $$AAOD_{\text{nd},\lambda} = (1 - \omega_{\text{nd},\lambda})AOD_{\text{nd},\lambda} \,. \tag{14}$$

We can assume that the light-absorbing features of the non-dust part of the aerosol plume are caused primarily by BC. It has been shown that BC is not an ideal light absorber, i.e., $\omega_{\text{BC},\lambda} \neq 0$, (*Bond and Bergstrom*, 2006; *Bond et al.*, 2013). **Thus**, we need to account for the SSA of BC to obtain the BC-related AAOD as:

$$AAOD_{\text{BC},\lambda} = AOD_{\text{nd},\lambda}(1 - \omega_{\text{nd},\lambda})(1 - \omega_{\text{BC},\lambda}) = AAOD_{\text{nd},\lambda}(1 - \omega_{\text{
[revised manuscript text omitted]

---

## Author Comment (AC4) · 7 Jan 2019

Interactive comments on "Technical note: Absorption aerosol optical depth components from AERONET observations of mixed dust plumes" by Sung-Kyun Shin et al.

Referee comments are noted in black. Our replies are given in blue.

We would like to thank all Referees for their constructive comments. Please find our point-by-point replies below. We have also attached a revised version of the manuscript with all changes marked.

**A. KOLGOTIN**

Could authors give more explanations or a reference to "the mixing rule" that is used to derive Eq. (12) in the line # 15?

In this study total SSA of dusty aerosol mixture was calculated by using the extinction-related dust ratio $\chi$ as defined in equation (10) and (11). $\chi$ determines the contribution of optical depth of dust and non-dust aerosols to the total optical depth of the mixed-dust plume.

Equation (10) and (11) could also be expressed as:

$$\chi_d = \frac{AOD_d}{AOD_d + AOD_{nd}}, \ \chi_{nd} = \frac{AOD_{nd}}{AOD_d + AOD_{nd}}$$

The $AOD_d$ and $AOD_{nd}$ denote aerosol optical depth of dust and non-dust particles, respectively. The SSA of total aerosol then could be expressed as a mixing of SSA for dust and SSA non-dust as given in equation (12).

*Noh et al.* (2014, 2016) suggested a weight factor that determines the contribution of the pure dust and non-dust part to the total optical signals (as we defined as the extinction-related dust ratio, $\chi_d$ and $\chi_{nd}$) for a retrieval of the SSA of the mixed-dust plume.

*Noh et al.* (2016) also reported that the distribution of SSA of mixed aerosols for each aerosol layers are nearly similar to the values of columnar-integrated SSA, which is calculated by adding the SSA of total mixed aerosol for each layer.

Noh, Y. M., Lee, K., Kim, K., Shin, S.-K., Müller, D., Shin, D. H.: Influence of the vertical absorption profile of mixed Asian dust plumes on aerosol direct radiative forcing over East Asia, Atmos. Env., 138, 191-204, https://doi.org/10.1016/j.atmosenv.2016.04.044, 2016.

Noh, Y. M.: Single-scattering albedo profiling of mixed Asian dust plumes with multiwavelength Raman lidar, Atmos. Env., 95, 305-317, https://doi.org/10.1016/j.atmosenv.2014.06.028, 2014.

Sorry, Eq. (15) in the line #25 is not obvious as well.

AERONET provides absorption aerosol optical depth (AAOD). Knowledge of SSA allows us to determine the fraction of AOD related to light absorption, AAOD, as:

$$AAOD = (1-\omega) \ AOD.$$

In our study, we separate dust AOD ($AOD_d$) and non-dust AOD ($AOD_{nd}$). Then we also calculate the single scattering albedo for non-dust particles. We then estimate the $AAOD_{nd}$ by using $AOD_{nd}$ and SSA of non-dust particles.

$AAOD_{nd}$ should be equivalent to $AAOD_{bc}$, if the entire light-absorbing properties of non-dust particles classified in this study came from BC (if SSA of BC is equal to 0). However, SSA

of BC is actually not zero. In order to take account of this consideration in the retrieval of AAOD for BC, we add the term $AAOD_{nd}(1-\omega_{BC})$ for this consideration in the equation.

[revised manuscript text omitted]
_d = \frac{(\delta - \delta_{nd})(1 + \delta_d)}{(\delta_d - \delta_{nd})(1 + \delta)} \tag{2}$$

15 and

$$R_{nd} = 1 - R_d. \tag{3}$$

Here, $\delta_d$ and $\delta_{nd}$ indicate $\delta$ of dust and non-dust particles, respectively. Their values can be determined from lidar measurements (*Burton et al.*, 2014; *Freudenthaler et al.*, 2009) or from AERONET observations representative for pure mineral dust (*Shin et al.*, 2018). At the standard lidar wavelength of 532 nm, typical values are $\delta_d = 0.33$ and $\delta_{nd} = 0.02$ (*Freudenthaler et al.*,
20 2009; *Burton et al.*, 2014). *Shin et al.* (2018) recently discussed AERONET-derived $\delta_d$ for mineral dust from different source regions.  **authors** conclude that in general, values of $\delta$ at 870 and 1020 nm from the AERONET version 3 inversion product seem to be most reliable . **Their finding is based on values found in literature that reports** on lidar observations of mineral dust. We consequently apply the aerosol-type separation procedure to AERONET measurements at 1020 nm.  **We used** values of $\delta_d = 0.30$ ($\delta_d = 0.31$) for mixed Asian (Saharan) dust plumes (*Shin et al.*,
25 2018) and $\delta_{nd} = 0.02$. **The latter value has been obtained from the analysis of $\delta$ derived at AERONET stations dominated by biomass-burning aerosols, analogous to the dust-focused study of *Shin et al.* (2018).** When $\delta$ was lower than $\delta_{nd}$ or higher than $\delta_d$, $R_d$ was set to 0 or 1, respectively.

The ratios $R_d$ and $R_{nd}$ obtained from using $\delta$ refer to the lidar measurements in the backscatter direction (i.e. the scattering angle of 180°) and allow for inferring the dust-related backscatter coefficient $\beta_d$ as:

30 $$\beta_d = \beta R_d. \tag{4}$$

This approach needs to be refined so that it can be also applied to sun/sky photometer measurements which provide information on total light extinction, i.e. AOD is the height integral of the extinction coefficient $\alpha$, rather than the backscatter

coefficient. For a single aerosol layer of depth $h$, it can be expressed as $AOD = \alpha h$. The extinction coefficient is connected to $\beta$ through the lidar ratio $S = \alpha/\beta$. Consequently, dust AOD can be expressed as:

$$AOD_d = S_d \beta_d h. \tag{5}$$

The use of Eq. (5) for the total aerosol and the dust fraction together with Eq. (4) leads to the dust and non-dust AOD as:

$$AOD_d = AOD \times R_d \times \frac{S_d}{S} \tag{6}$$

and

[revised manuscript text omitted]

---

## Author Comment (AC5) · 7 Jan 2019

Interactive comments on "Technical note: Absorption aerosol optical depth components from AERONET observations of mixed dust plumes" by Sung-Kyun Shin et al.

Referee comments are noted in black. Our replies are given in blue.

We would like to thank all Referees for their constructive comments. Please find our point-by-point replies below. We have also attached a revised version of the manuscript with all changes marked.

**Anonymous Referee #2 (A. Lyapustin)**

P2, Ln15: "Russell et al. (2010) utilized AERONET-retrieved SSA, AAOD, and absorption Ångström exponent (AAE) as indicator to separate the contributions of BC, organic matter (OM), and mineral dust to the absorbing aerosol fraction." That is incorrect: This work suggested a preferred classification scheme, not a "separation".

Thank you for the clarification. We have revised our statement to:

*"Russell et al. (2010) utilized AERONET-retrieved SSA, AAOD, and absorption Angstrom exponent (AAE) as indicator to classify observations with respect to the contributions of BC, organic matter (OM), and mineral dust to the absorbing aerosol fraction."*

P2, L21: "…and to infer the fine or coarse mode fraction in the aerosol size distribution" What does it mean?

We thank the reviewer for this comment. Schuster et al. (2006) investigate the relationship between the Angstrom exponent and the mode (fine and coarse) parameters of bimodal aerosol size distributions.

For clarification, we have revised the statement to:

*"For instance, the Ångström exponent (AE or å, Ångström 1964) as inferred from spectral AOD measurements gives qualitative information on aerosol size that can be used for aerosol-type classification: values greater than 2 indicate small particles such as biomass-burning smoke while values smaller than 1 indicate large particles like sea salt and mineral dust. Schuster et al. (2006) found that the variation of the Ångström exponent is associated with bimodal aerosol size distributions. The authors focused on the fine or coarse fraction of aerosols."*

P3, L10: "The uncertainty in SSA is expected to be of the order of 0.03 (Holben et al., 1998)." The reference is incorrect.

We have changed the reference to Dubovik et al. (2000).

Eq. (4) in my view is incorrect. The correct equation for the measured backscattering should be something like Beta = f_dust*Beta_dust + f_nd*Beta_nd, where f_dust=AOD_dust/AOD, AOD=AOD_dust+AOD_nd. If phase functions of dust and non-dust are very different, and they usually are because of the particle size difference, then Eq. (4) is not correct.

I may be wrong as I did not work with the lidars before. But if I am correct, then this error (or assumption) propagates in all derivations below. I suggest a major revision with authors addressing this key point first.

We thank the Referee for his concerns. We are confident that Eq. (4) is correct as it describes the dust part only. We would like to point out that Eqs. (2) – (5) refer to lidar measurements only, i.e. to measurements at 180 degree backscatter direction (only one point of the phase function). The methodology used here has been established by *Shimizu et al.* (2004) and

*Tesche et al.* (2009b) for lidar measurements that include the particle linear depolarisation ratio. The core of our paper is to adapt this methodology to column-integrated AERONET measurements. This requires a transition from the backscatter coefficient to AOD that is facilitated by Eqs. (5) and (6).

[revised manuscript text omitted]

---

## Author Comment (AC6) · 7 Jan 2019

Interactive comments on "Technical note: Absorption aerosol optical depth components from AERONET observations of mixed dust plumes" by Sung-Kyun Shin et al.

Referee comments are noted in black. Our replies are given in blue.

We would like to thank all Referees for their constructive comments. Please find our point-by-point replies below. We have also attached a revised version of the manuscript with all changes marked.

**Anonymous Referee #5**

Manuscript presents a new technique to distinguish between contributions of dust and non-dust aerosols to the total aerosol optical depth (AOD) measured by AERONET. The approach is based on utilizing the particle linear depolarization ratio (PLDR) which is now available as one of aerosol products in AERONET Version3. Due to PLDR sensitivity to aerosol particle shape, the dust/non-dust separation is based on the particles shape differences rather than on difference in aerosol size. This, for example, allows excluding contribution of coarse spherical particles from dust AOD component.

I believe that the subject of manuscript is in scope of AMT, it is very well written, and presented results are interesting. I recommend it to be published with minor revisions.

Comments

1. My main comment is regarding Eq. (15). It is not clear how it was derived and what model of non-dust aerosol component was assumed. Equation suggests that black carbon (BC) is not the only absorber in non-dust component (both BC and non-dust single scattering albedos are present in (15)). However if the second absorber is present its properties need to be described. In addition, the right side of the equation suggests that AAODnd should be equivalent to BC extinction AOD, which is not obvious without knowing what model of non-dust component was employed. I was able to derive the equation assuming the presence of the second very low absorbing component in non-dust component mixture. However, it still not clear how it was done by authors. I suggest to include discussion on how the Eq. (15) was derived and what assumptions were used.

We thank the review for this comment. After extracting the influence of the light absorbing characteristics of mineral dust on AAOD we are left with the AAOD related to non-dust particles. This could include black carbon, organic particulate, and sulphate with varying light absorbing properties. We now assume that the light-absorbing properties of those aerosol components are dominated by the contribution of BC. As the reviewer pointed out, $AAOD_{nd}$ should be equivalent to $AAOD_{BC}$ in the sense of that absorbing properties of the non-dust aerosol are caused solely by BC. However, laboratory measurements have shown that BC is not an ideal absorber ($\omega_{BC} \neq 0$) which needs to be reflected in the $AAOD_{BC}$ retrieval. We thus added the term for this consideration to Eq. (15).

2. Page 3, between lines 5 and 10 authors write: "The AERONET inversion is performed for measurements with a 440-nm AOD larger than 0.4". Actually inversion is performed for all the values of AOD, but only inversions for AOD (440)>0.4 are included in Level 2 product.

Thank you for the clarification. We have changed the statement to:

*"The level 2 product available from the AERONET portal includes inversion results for measurements with a 440-nm AOD larger than 0.4 (Dubovik et al., 2006). The AERONET inversion uses…"*

3. Page 3, line 10. For reference on uncertainty in SSA retrievals is more appropriate to use Dubovik (2000) uncertainty paper.

We have changed the reference to Dubovik et al. (2000).

4. Page 5. "Re-arranging Eq. (11)". I think it should be Eq. (12).

Changed.

5. Page 8. "using Eqs. (6) and (6),". Probably typo.

This has been corrected to Eqs. (6) and (7).

6. Page 8. Why analysis of AAOD (BC) was conducted at all four AERONET wavelength? The earlier discussion suggested using just 1020 nm PLDR product as being more reliable.

As we stated in this manuscript, the strongest correlation exists between the AERONET-derived PLDR at 1020 nm and the lidar-derived PLDR at 532 nm. We hence use this wavelength for the initial separation between the contributions of dust and non-dust aerosols to the dusty aerosol mixture. Once this separation is done with PLDR at 1020 nm, we can apply this information to AOD at other wavelengths using the Ångström exponent for pure dust in Eq. (8). This means that $AAOD_{BC}$ at wavelength smaller than 1020 nm has been obtained in a different way to the one at 1020 nm, i.e. without using PLDRs at the respective wavelengths. We present $AAOD_{BC}$ at all four standard AERONET wavelengths to illustrate the full potential of our methodology.

7. Figure 7 caption. Upper and lower panels should be probable left and right panels.

Changed.

8. You may consider using AERONET SDA product to compare to dust AOD in addition to coarse mode AOD inferred from particle size distribution retrievals.

We thank the reviewer for this comment. While following up in this recommendation, we realised that such a comparison is not straightforward. The SDA product gives coarse- and fine-mode AOD only at 500 nm. While our currently considered products could be transferred to this wavelength with the help of the Angstrom exponent between 440 and 670 nm, we also found that the SDA retrieval provides results at times different to the products we have currently included in our study. Because of these ambiguities we do not consider a comparison between SDA coarse mode AOD and dust AOD in the revised version of the manuscript.

[revised manuscript text omitted]
_d = \frac{(\delta - \delta_{nd})(1 + \delta_d)}{(\delta_d - \delta_{nd})(1 + \delta)} \tag{2}$$

and

$$R_{nd} = 1 - R_d. \tag{3}$$

Here, $\delta_d$ and $\delta_{nd}$ indicate $\delta$ of dust and non-dust particles, respectively. Their values can be determined from lidar measurements (*Burton et al.*, 2014; *Freudenthaler et al.*, 2009) or from AERONET observations representative for pure mineral dust (*Shin et al.*, 2018). At the standard lidar wavelength of 532 nm, typical values are $\delta_d = 0.33$ and $\delta_{nd} = 0.02$ (*Freudenthaler et al.*, 2009; *Burton et al.*, 2014). *Shin et al.* (2018) recently discussed AERONET-derived $\delta_d$ for mineral dust from different source regions. The **authors** conclude that in general, values of $\delta$ at 870 and 1020 nm from the AERONET version 3 inversion product seem to be most reliable  **Their finding is based on values found in literature that reports** on lidar observations of mineral dust. We consequently apply the aerosol-type separation procedure to AERONET measurements at 1020 nm.  **We used** values of $\delta_d = 0.30$ ($\delta_d = 0.31$) for mixed Asian (Saharan) dust plumes (*Shin et al.*, 2018) and $\delta_{nd} = 0.02$. **The latter value has been obtained from the analysis of $\delta$ derived at AERONET stations dominated by biomass-burning aerosols, analogous to the dust-focused study of *Shin et al.* (2018).** When $\delta$ was lower than $\delta_{nd}$ or higher than $\delta_d$, $R_d$ was set to 0 or 1, respectively.

The ratios $R_d$ and $R_{nd}$ obtained from using $\delta$ refer to the lidar measurements in the backscatter direction (i.e. the scattering angle of 180°) and allow for inferring the dust-related backscatter coefficient $\beta_d$ as:

$$\beta_d = \beta R_d. \tag{4}$$

This approach needs to be refined so that it can be also applied to sun/sky photometer measurements which provide information on total light extinction, i.e. AOD is the height integral of the extinction coefficient $\alpha$, rather than the backscatter

coefficient. For a single aerosol layer of depth $h$, it can be expressed as $AOD = \alpha h$. The extinction coefficient is connected to $\beta$ through the lidar ratio $S = \alpha/\beta$. Consequently, dust AOD can be expressed as:

$$AOD_d = S_d \beta_d h. \tag{5}$$

The use of Eq. (5) for the total aerosol and the dust fraction together with Eq. (4) leads to the dust and non-dust AOD as:

$$AOD_d = AOD \times R_d \times \frac{S_d}{S} \tag{6}$$

and

[revised manuscript text omitted]

---

## Author Comment (AC7) · 7 Jan 2019

Please see the uploaded file for our replies to your comments.

Please also note the supplement to this comment:
https://www.atmos-meas-tech-discuss.net/amt-2018-311/amt-2018-311-AC7-supplement.pdf
* * *

---

## Author Comment (AC2)

Interactive comments on "Technical note: Absorption aerosol optical depth components from AERONET observations of mixed dust plumes" by Sung-Kyun Shin et al.

Referee comments are noted in black. Our replies are given in blue.

We would like to thank all Referees for their constructive comments. Please find our point-by-point replies below. We have also attached a revised version of the manuscript with all changes marked.

**Anonymous Referee #4**

General comments

The analysis of the AAOD components of mixed dust plumes is an important topic for scientific community and the proposed method to apply a lidar technique to the AERONET v3 inverse products could be very interesting. However, the authors should clarify key issues to make robust and rigorous the approach presented in this note. Besides, major revision of the overall presentation should be properly addressed before the publication.

Specific comments

1) The level 2.0 assures the quality level of direct and inverse AERONET products. What about the level (1.0, 1.5 or 2.0) of the AERONET data used in this study?

Only level 2.0 data have been considered in this study. This is stated repeatedly in the text, e.g. at the end of the Introduction, at the end of Section 2.1, and in the beginning of the Summary.

2) Have the authors used the AERONET's recommended loading constraint (AOD>0.4 at 440nm) for inverse products?

Yes. We have only considered level 2.0 inversion products, which are only provided for AOD(440 nm) > 0.4.

3) Shin et al., 2018 reported specific conditions for PLDR in case of pure mineral dust: "To select observations representative of pure mineral dust conditions, only AERONET data with a 440/870 nm Ångström exponent below 0.4 and a fine-mode fraction below 0.10 have been selected in this study." The authors should explain how they overcome both these conditions considering the values of PLDR reported in Shin et al., 2018 in different FMF and Ångström exponent domains.

The lidar-based aerosol-type-separation method that forms the foundation of our study requires reference values of pure (i.e. unmixed) aerosol types. We use the values of *Shin et al.* (2018) as reference for pure dust from different source regions. Observations that don't meet the constraints listed above are automatically considered as mixed dust. Hence, there is no need to overcome the conditions used in *Shin et al.* (2018).

4) The methodology for the retrieval of AAOD of dust and BC components is based on Equation (6). From the description of methodology, the authors assumed the same single aerosol layer of depth h for both aerosol mixing and dust. Furthermore, the integrated-values of the extinction coefficient for the mixed plume and the dust component were solved by assuming the extinction coefficient constant in the layer. The AOD AERONET product represents the integration of the vertically varying extinction coefficient in the entire atmospheric column. The authors should explain how the integration domain of the total columnar mixing aerosol (AOD) can be limited to a single layer in which the aerosol mixing

and the dust component are limited. Furthermore, the assumption of vertically non-varying extinction coefficient should be in-depth explained.

In this study we cannot resolve details on aerosol layering as the parameters provided by AERONET refer to the columnar integral, e.g., AOD and single-scattering albedo for the entire atmospheric column. For this reason we have to assume that different types of aerosol are mixed in the total atmospheric column. We then can separate the dust and non-dust AOD in the total AOD.

As the reviewer pointed out, the interpretation of layer mean values can be rather misleading when multiple aerosol layers are being present. Lidar allows for capturing the vertical variation of optical parameters in these complex aerosol mixtures. However, our approach suggests a way to obtain comparable information from AERONET data when more costly lidar measurements are not available. *Noh et al.* (2016) have shown that layer-mean aerosol parameters as obtained from observations with lidar and by AERONET instruments are generally in very good agreement for the majority of considered cases.

Noh, Y. M., Lee, K., Kim, K., Shin, S.-K., Müller, D., Shin, D. H., Influence of the vertical absorption profile of mixed Asian dust plumes on aerosol direct radiative forcing over East Asia, Atmos. Env., 138, 191-204, https://doi.org/10.1016/j.atmosenv.2016.04.044, 2016.

Technical corrections

p. 3 l. 4 Please, explicit the amount of AERONET bands (not 'several')

We used this fuzzy formulation as some instruments operate at more or different wavelengths than others. We have now refined this statement to:

*"AERONET instruments measure AOD at wavelengths from 340 nm to 1640 nm always including observations at 440, 670, 870, and 1020 nm.*

p. 4 l. 11 Probably, a bracket is missing

We have added the closing bracket.

p. 8 l. 9 'Eqs (6) and (6)'. Please, control the reference to equations.

This has been corrected to Eqs. (6) and (7).

p. 8 l. 31 'therefoe'

Changed

(2) effect of uncertainty in retrieving the aerosol characteristics by the standard AERONET algorithm on the AAOD retrievals, based on this method

In order to calculate the contribution of $AAOD_{BC}$ we used AERONET-retrieved level 2.0 AOD and SSA. The AOD uncertainty is estimated as 0.01 to 0.02 depending on wavelength in the absence of cloud contamination. The uncertainty in SSA is expected to be of the order of 0.03. The uncertainty we obtain in our study ranges within the standard deviation of the daily mean $AAOD_{BC}$. We hence believe that the values of $AAOD_{BC}$ obtained with our approach are reasonable. In any case, the choice of reference values, i.e. PLDR of dust and non-dust and most importantly the choice of SSA of the absorbing aerosol component, has a stronger effect on the results than the uncertainties of the AERONET products.

All technical shortcomings, noted by me, have already been indicated by other reviewers and are hoped to be accounted for in the final version of the manuscript.

We have addressed all of them.

[revised manuscript text omitted]

Re-arranging Eq. (12) gives the SSA related to non-dust particles

$$\omega_{\text{nd},\lambda} = \frac{\omega_\lambda - \chi_{\text{d},\lambda}\omega_{\text{d},\lambda}}{\chi_{\text{nd},\lambda}}.$$ (13)

The spectral SSA for pure dust particles is taken from the literature (see Table 1). The non-dust fraction to AAOD can now be derived as

5  $$AAOD_{\text{nd},\lambda} = (1 - \omega_{\text{nd},\lambda})AOD_{\text{nd},\lambda}.$$ (14)

We can assume that the light-absorbing features of the non-dust part of the aerosol plume are caused primarily by BC. **I**t has been shown that BC is not an ideal light absorber, i.e., $\omega_{\text{BC},\lambda} \neq 0$, (*Bond and Bergstrom*, 2006; *Bond et al.*, 2013). **Thus**, we need to account for the SSA of BC to obtain the BC-related AAOD as:

$$AAOD_{\text{BC},\lambda} = AOD_{\text{nd},\lambda}(1 - \omega_{\text{nd},\lambda})(1 - \omega_{\text{BC},\lambda}) = AAOD_{\text{nd},\lambda}(1 - \omega_{\text{BC},\lambda}).$$ (15)

[revised manuscript text omitted]